# Ruminal Fluid Transplantation Accelerates Rumen Microbial Remodeling and Improves Feed Efficiency in Yaks

**DOI:** 10.3390/microorganisms11081964

**Published:** 2023-07-31

**Authors:** Yan Li, Yingkui Yang, Shatuo Chai, Kaiyue Pang, Xun Wang, Linpeng Xu, Zheng Chen, Yumin Li, Tanqin Dong, Weihua Huang, Shujie Liu, Shuxiang Wang

**Affiliations:** 1Qinghai Academy of Animal Husbandry, Veterinary Sciences in Qinghai University, Xining 810016, China; zgsjhhxxwl@163.com (Y.L.); yykui@qhu.edu.cn (Y.Y.); chaishatuo@163.com (S.C.); pky0425@163.com (K.P.); wangxun513@163.com (X.W.); xlp0198@126.com (L.X.); czcjsno1@163.com (Z.C.); LYM395062029@outlook.com (Y.L.); dongtanqin2022@163.com (T.D.); 17609787857@163.com (W.H.); mkylshj@126.com (S.L.); 2Key Laboratory of Plateau Grazing Animal Nutrition and Feed Science of Qinghai Province, Xining 810016, China; 3Yak Engineering Technology Research Center of Qinghai Province, Xining 810016, China

**Keywords:** ruminal fluid transplantation, yak, feed efficiency, ruminal microorganisms, house feeding

## Abstract

**Simple Summary:**

The yak is a unique species found on the Tibetan plateau, adapted to the high-altitude, low-oxygen environment, providing meat, milk, and other resources to local herders, and is an indispensable means of production and livelihood for herders on the Tibetan plateau and a major source of economic income for Tibetan herders. A change in diet leads to indigestion and anorexia during the acclimatization phase, which can lead to changes in dry matter intake and daily weight gain in the early part of the housing period. The rumen microbiota will be reshaped. Ruminal fluid transplantation can reshape the rumen microbiota of the recipient animals, which can rapidly restore the microbiota to a new homeostasis. Considering the special characteristics of yaks on the Tibetan plateau, the dynamic seasonal changes in pasture and the trend of changing from natural grazing to confinement farming, this study focused on the changes in rumen flora due to dietary changes and the improvement of rumen fluid transplantation on the remodeling of yak flora and production performance when yak breeding patterns were changed.

**Abstract:**

A relatively stable microbial ecological balance system in the rumen plays an important role in rumen environment stability and ruminant health maintenance. No studies have reported how rumen fluid transplantation (RFT) affects the composition of rumen microorganisms and yak growth performance. In this experiment, we transplanted fresh rumen fluid adapted to house-feeding yaks to yaks transitioned from natural pastures to house-feeding periods to investigate the effects of rumen fluid transplantation on rumen microbial community regulation and production performance. Twenty yaks were randomly divided into the control group (CON; n = 10) and the rumen fluid transplantation group (RT; n = 10). Ten yaks that had been adapted to stall fattening feed in one month were selected as the rumen fluid donor group to provide fresh rumen fluid. Ruminal fluid transplantation trials were conducted on the 1st, 3rd, and 5th. Overall, 1 L of ruminal fluid was transplanted to each yak in the RT and CON group. The formal trial then began with both groups fed the same diet. After this, growth performance was measured, rumen fluid was collected, and rumen microbial composition was compared using 16s rRNA sequencing data. The results showed that rumen fluid transplantation had no significant effect on yak total weight gain or daily weight gain (*p* > 0.05), and feed efficiency was higher in the RT group than in the CON group at 3 months (treatment × month: *p* < 0.01). Ruminal fluid transplantation significantly affected rumen alpha diversity (*p* < 0.05). Up to day 60, the RT group had significantly higher OTU numbers, Shannon diversity, and Simpson homogeneity than the CON group. Principal coordinate analysis showed that the rumen microbiota differed significantly on days 4 and 7 (*p* < 0.05). Bacteroidota, Firmicutes, Proteobacteria, and Spirochaetes were the most abundant phyla in the rumen. The relative abundances of Bacteroidota, Proteobacteria, and Spirochaetes were lower in the RT group than in the CON group, with a decrease observed in Bacteroidota in the RT group on days 7 and 28 after rumen fluid transplantation (*p* = 0.013), while Proteobacteria showed a decreasing trend in the CON group and an increasing trend in RT; however, this was only at day 4 (*p* = 0.019). The relative abundance of Firmicutes was significantly higher in the RT group than in the CON group on days 4, 7, and 28 (*p* = 0.001). *Prevotella* and *Rikenellaceae_RC9_gut_group* were the predominant genera. In conclusion, our findings suggest that rumen fluid transplantation improves yak growth performance and rumen microbial reshaping. The findings of this study provide new insights into yak microbial community transplantation and a reference for improving feed efficiency in the yak industry.

## 1. Introduction

Yak is a unique species found on the Tibetan Plateau that has adapted to the high-altitude, low-oxygen environment by providing meat, milk, and other resources to local herders. It is the largest ruminant at high altitudes and the primary source of economic income for Tibetan herders [1]. Due to the harsh environment in high-altitude areas, the dynamic seasonal changes in pastures cause an imbalance in grass nutrient supply, especially in winter under the traditional grazing mode, which eventually has a serious impact on yak performance and even leads to death [2,3], which is obviously unable to meet the requirements of yak quality and efficiency improvement. With the continued development of ecological animal husbandry, the number of yak breeders in Qinghai Province is expected to reach 5 million by 2025, with 1.5 million of those yaks being house-fed. Therefore, it has become inevitable for some yaks to switch from natural-based production to house-feeding, which positively impacts the development of the yak industry [4]. Increasing the house-feeding scale and switching from natural pasture to a high-grain diet causes indigestion and anorexia in yaks in the adaptation stage. This can change the dry matter intake and daily weight gain of yaks during the early stage of the housing period [5]. Furthermore, this directly affects the production performance of yaks and becomes a problem that affects the high-quality development in the industry.

Yaks have a unique rumen microbiota that enables them to survive in harsh environments [6], which is widely regarded as the main cause of the variation in feed conversion productivity in ruminants [7,8]. Previous studies have shown that the microbiota living in the gastrointestinal tract plays an irreplaceable role in host adaptation to the environment, passive immunity, nutrient absorption, and metabolism in ruminants [9,10]. Currently, there are more studies on how to regulate rumen microbiota to influence metabolic disorders and growth performance [11]. Studies in mice have found a strong correlation between organismal metabolic disorders and microbial community composition. They have also demonstrated that fecal microbial transplants effectively restore gastrointestinal function [12,13]. Gastric fluid transplantation can reshape the structure of the animal rumen microbiota and can quickly restore the microbiota of the recipient animal to homeostasis, resulting in improved organism health and productivity [14]. From a microbial ecological perspective, young rumen ecosystems are easier to repair because their microbial communities are simpler and less resistant to colonization than those in adult ruminants [15]. The study found that recipient cows experienced rumen microbial remodeling, increased dry matter and feeding frequency in periparturient cows, and decreased dry matter intake in dyspeptic cows and rumen protozoa numbers. This demonstrates that rumen fluid transplantation effectively regulates the rumen microbiota [16,17]. In a mid-20th century study, calves were inoculated with rumen fluid or rumen contents from adult cattle, which showed that microbial inoculation accelerated the establishment of rumen protozoa in calves [18]. Supplementing rumen fluids can promote rumen maturation, improve rumen papilla growth, and dry matter intake, and affect the composition of the rumen microbial community, according to studies on rumen microbial colonization and rumen development [19]. However, there has been little research into how rumen fluid transplantation affects rumen fermentation and microbial community dynamics in yaks adapted to high grain ratios.

Considering the unique characteristics of yaks on the Tibetan Plateau, the dynamic seasonal changes in pastures, and the trend of natural grazing in the house-feeding model, this study focused on the changes in rumen flora due to dietary changes and the effect of rumen fluid transplantation on the reshaping of rumen flora and production performance. Therefore, in this study, we obtained fresh rumen fluid from yaks that had adapted to the house-feeding diet and inoculated it three times into the experimental group to remodel the microbial community. We then used high-throughput sequencing (by 16s rRNA sequencing) to understand the dynamic changes in rumen microorganisms and the production performance of the transplanted group over 90 days. This study is the first to report the effect of rumen fluid transplantation on yak growth performance and rumen microbiota reshaping.

## 2. Materials and Methods

### 2.1. Animals, Diets, and Trial Design

The trial was conducted in December 2021 at Zeku County National Agricultural Industrial Park, Huangnan Prefecture, Qinghai Province, China. The Institutional Animal Care and Use Committee of Qinghai University approved all procedures in this study.

Twenty 6-month-old healthy male yaks (weight 57.2 ± 7.8 kg) were selected from an alpine meadow pasture at an altitude of approximately 3400 m, transferred to the farm, and randomly divided them into a control group (CON; n = 10) and a rumen fluid transplantation group (RT; n = 10). Separate single-pen rearing was performed in two pens using the same rearing environment and feeding method and all cattle were earmarked. In addition, ten yaks that had been acclimated to captive breeding feed and grazed in the same alpine meadow before captive breeding were selected as the rumen fluid donor group to provide fresh rumen fluid to the rumen fluid transplantation group.

The trial was divided into two parts, the rumen fluid transplantation trial and a formal trial. The first phase lasted 5 d, and the second lasted 90 d. The rumen fluid transplantation trial started on the 1st day of the arrival of the yaks to the farm. First, 3.5 L of rumen fluid was collected from the donor group of yaks using a gastric tube sampler, filtered through four layers of sterile gauze, and mixed into a thermos flask for backup. Then, using a gastric tube sampler, approximately 340 mL of fresh rumen fluid was transplanted to the RT group of yaks. All the above steps were completed in the morning before feeding and repeated on days 3 and 5 of rumen fluid transplantation to ensure that each yak in the RT group was transplanted with 1 L of fresh rumen fluid.

After the official trial began, the health and behavior of the yaks were monitored daily. The CON and RT groups were given enough oat hay and concentrate. In contrast, the yaks were given free access to water and fed twice daily at 09:00 and 17:00. Each morning, the amount of oat hay and concentrate was left over from the previous day. The amount of input for the day was recorded each morning to determine the daily feed intake of the yak-reading cattle. The composition of the concentrate is listed in Table 1. The main components of oat grass were (DM basis): 6.1% crude protein (CP), 3.15% ether extract (EE), 49.8% neutral detergent fiber (NDF), 31.3% acid detergent fiber (ADF), 0.5% Ca, and 0.08% P. Daily intake is shown in (Appendix A).

### 2.2. Sample Collection and Measurement

The initial feed offer and refusal of the yaks were recorded daily. The yaks were weighed before feeding on the morning of the 1st day of the formal trial, and then measured monthly until the trial ended.

Yaks in the CON and RT groups received rumen fluid before being transferred to the farm for the trial, using a bendable oral gastric tube with a metal filter that was pre-cleaned by rinsing with clean, warm water. The first 50 mL of rumen fluid was discarded to avoid saliva contamination. Finally, each yak had a 50 mL rumen fluid sample collected and filtered through four layers of gauze. The samples were divided into 15 mL sterile centrifuge tubes for microbial analysis and stored in liquid nitrogen. The procedure described above was repeated for rumen fluid samples on days 4, 7, 14, 28, 60, and 90 after the formal experiment began. On days 4, 28, and 90 after the experiment started, blood was collected from the caudal vein on a fasting basis, centrifuged after standing, and 2 mL of supernatant in liquid nitrogen for 30 days. Blood samples were centrifuged once more, and the supernatant was stored in liquid nitrogen. The parameters were measured using a fully automated biochemical instrument (AU5831, Belman Coulter, Pasadena, CA, USA.

### 2.3. 16s rRNA Gene Amplification and MiSeq Sequencing

The CTAB (Hexadecyl trimethyl ammonium Bromide) method was used to extract microbial DNA from the rumen fluid samples. First, 1.0% agarose gel electrophoresis was used to determine the concentration and purity of DNA. Depending on the concentration, the DNA was diluted to 1 ng/μL with sterile water. The extracted DNA was then used for polymerase chain reaction (PCR) amplification to amplify different regions of the 16s rRNA gene (16sV3–V4) using specific primers 515F (5′-GTGCCAGCMGCCGCGG-3′) and 806R (5′-GTGCCAGCMGCCGCGG-3′) with barcodes. PCR was performed using a 25-μL amplification system, 5 μmol/L of upstream and downstream primers, and approximately 5 ng of template DNA. The PCR amplification conditions were as follows: pre-denaturation at 94 °C for 5 min, followed by 30 s of denaturation at 94 °C, annealing at 50 °C for 30 s, and extension at 72 °C for 60 s for 30 cycles, followed by extension at 72 °C for 7 min. Then, 1.0% agarose gel electrophoresis was used to detect the PCR amplification products, and the recovered products were purified using the MinElute Gel Extraction Kit (Qiagen, Düsseldorf, Germany). Equal amounts of purified amplicons were pooled together to construct a paired-end sequencing library. PCR amplification, mixing and purification of PCR products, library construction, and up-sequencing processes were performed by Ltd. (Beijing, China) using the platform (Illumina NovaSeq6000, San Diego, CA, USA) for sequencing according to the standard protocol.

### 2.4. Sequence and Rumen Microflora Processing

Sequences obtained from the Illumina NovaSeq6000 platform were processed using the open-source software pipeline QIIME (Quantitative Insights into Microbial Ecology) version 1.8.0-dev [20]. Briefly, (1) studies with an average quality score of no less than 20 and no shorter than 50 bp were retained; (2) reads with exact barcode matches, two nucleotide mismatches in a single match, and unclear characters were discarded; (3) only sequences overlapping by more than 10 bp were assembled based on their overlapping sequences. Reads that were unable to be assembled were excluded. UCLUST (version 7.1, http://drive5.com/uparse/, 5 June 2023) and chimeric sequences were identified and removed using UCHIME [21]. The most stable sequences within each OTU from a specific library bacterium were designated as “representative sequences”, and they were compared to the Silvabacterial database (version 119) [22] and the SILVA using PYNAST ArchaeadDatabase [20] with the default parameters set by Qiime. Community richness and diversity were analyzed using measurements such as Chao1, Shannon, PD-whole-tree, Observed-species, principal coordinates analysis based on weighted UniFrac distances principal coordinate analysis (PCoA), and analysis of molecular variance based on weighted distances (AMOVA) were evaluated with the program method v.1.35.0 for sequences used to account for significant differences between samples.

### 2.5. Statistical Analysis

The linear mixed-effects modeling procedure of SPSS Statistics (version 25.0; IBM Corp., Armonk, NY, USA) was used to analyze the growth performance metrics. Treatment (CON or RT), number of days, and their interactions were considered fixed factors. Yaks were treated as random effects, while days were treated as repeated measures. If the model indicated a significant difference in sampling time, the Tukey honest significant difference test (Rv.3.6.3) was applied for subsequent multiple comparisons. If the model revealed a significant difference between treatments (CON or RT), the two groups were compared at the same sampling time point using a *t*-test (R v.3.6.3), and the difference was considered statistically significant (*p* < 0.05). The diversity between samples was assessed as Hellinger-transformed Bray–Curtis differences and visualized using PCoA to investigate the effects of treatment and time on changes in microbial community structure. The Shapiro–Wilk test was used to evaluate the normal distribution of each variable of interest. Differences in alpha diversity indices and relative abundance of bacterial communities were analyzed using the non-parametric Scheirer–Ray Hare extension of the Kruskal–Wallis test. The PICRUSt 2 software was used to predict microbiota function and explore the differences between the two groups. Differences between the two groups were determined for level 2 of the Kyoto Encyclopedia of Genes and Genomes pathway.

## 3. Results

### 3.1. Effect of Rumen Fluid Transplantation on Growth Performance and Blood Metabolites in Yaks

At the beginning of the experiment, there was no significant difference in the average body weight of yaks in each group (CON: 59.8 ± 2.7; RT: 59.2 ± 2.6; least square means (LSM) ± standard error (SE) kg). The trend of increasing daily weight gain was more pronounced in the RT group than in the CON group throughout the treatment period (months: *p* < 0.001) (Figure 1B). At month 3, feed efficiency in the RT group was higher than that in the CON group (treatment × month: *p* < 0.01) (Figure 1C).

Throughout the experiment, rumen fluid transplantation had a significant effect on serum levels of Ca, GLU, TP, and CHO in yaks (Trt: *p* < 0.05; Day: *p* < 0.05) (Table 2). On Day 4, the RT group had significantly higher Ca (*p* = 0.014) and TP (*p* = 0.029) serum levels than in the CON group; on day 28, the RT group had significantly higher GLU (*p* = 0.027), CHO (*p* = 0.005), and TG (*p* = 0.039) serum levels than in the CON group. However, on day 90, yak serum Ca (*p* = 0.043) and TP (*p* = 0.027) levels were significantly higher in the RT group than in the CON group.

### 3.2. Effect of Rumen Fluid Transplantation on VFA in Yaks

Ruminal fluid transplantation had a significant effect on the levels of Isobutyrate and Butyrate in VFA of yaks throughout the experiment (Table 3). At day 4, Isobutyrate (*p* = 0.001) and Butyrate (*p* = 0.012) levels were significantly higher in the RT group than in the CON group. At day 28, Isobutyrate (*p* = 0.004) content was significantly higher in the RT group than in the CON group. At day 60, Isobutyrate (*p* = 0.001) content was significantly higher in the RT group than in the CON group.

### 3.3. Effect of Rumen Fluid Transplantation on Rumen Bacterial Diversity

Overall, 5,139,387 sequences from 60 samples were generated by amplicon sequencing. In all samples, 6606 OTU species were identified, 3281 OTU species in the CON group and 3325 OTU species in the RT group, with 798 genera of 56 bacterial families detected.

The treatment significantly affected alpha diversity in the rumen of yaks throughout the trial (treatment: *p* < 0.05). Until day 60, the RT group had significantly higher OTUs, Shannon diversity index, and Simpson homogeneity index than those in the CON group (Table 4). The first and second principal components contributed 28.29% and 16.76%, respectively, as revealed using PCoA analysis based on UniFrac. In addition, we found that samples within 4–7 d were significantly separated along axis 2, whereas these samples were clustered together after 7 d (Figure 2).

### 3.4. Effect of Rumen Fluid Transplantation on Changes in Bacterial Composition

At the phylum level, 56 phyla were identified, the dominant phyla comprising Bacteroidota, Firmicutes, Proteobacteria, and Spirochaetes accounting for more than 1% (Figure 3). Bacteroidota and Firmicutes were the dominant phyla accounting for 49.11% and 39.94% of the total phylum, respectively. The relative abundance of Bacteroidota, Proteobacteria, and Spirochaetes were lower in the RT group than in the CON group. A decreasing trend was observed in the RT group for Bacteroidota at days 7 and 28 after rumen fluid transplantation (Figure 4A; *p* = 0.013). In contrast, Proteobacteria had a decreasing trend which was observed in the CON group, and an increasing trend in RT; however, this was only at day 4 (Figure 4C; *p* = 0.019). Moreover, the relative abundance of Firmicutes was significantly higher in the RT group than in the CON group on days 4, 7, and 28 (Figure 4B; *p* = 0.001). At d 7 and 28, the F/B of the RT group was significantly higher than that of the CON group, while at d 60, the F/B of the CON group suddenly increased significantly more than that of the RT group.

At the genus level, 798 genera were identified in yak rumen samples, with 16 abundant genera >1% (Table 5). *Prevotella* (19.37%) and *Rikenellaceae_RC9_gut_group* (11.11%) were the most dominant genera. Compared to the CON group, the RT group had *Christensenellaceae_R-7_group* (*p* = 0.002), *Ruminococcaceae_NK4A214_group* (*p* < 0.001), *Prevotellaceae_UCG-004* (*p* < 0.01), *Prevotellaceae_UCG-003* (*p* = 0.011), *Oscillospira* (*p* < 0.001), and *selenomonas* (*p* = 0.034). On days 4 and 7 after rumen fluid transplantation in the RT group, *Christensenellaceae_R-7_group*, *Ruminococcaceae_NK4A214_group*, *Prevotellaceae_UCG-004*, *Prevotellaceae_UCG-003*, and *Selenomonas* were significantly increased; microbial differences were observed for *Prevotellaceae_UCG-004* and *Prevotellaceae_UCG-003* on day 14 after rumen fluid transplantation in the RT group. On day 28 after rumen fluid transplantation in the RT group, *Christensenellaceae_R-7_group*, *NK4A214_group*, and *Selenomonas* percentage were significantly increased, while at day 60 after rumen fluid transplantation in the RT group, the difference in *Prevotellaceae_UCG-003* was observed; *Oscillospira* was observed significantly on days 4, 7, 14, 28, 60, and 90 after rumen fluid transplantation. The interaction between treatment groups and sampling days significantly affected the percentage of *Rikenellaceae_RC9_gut_group* (*p* = 0.012), *Christensenellaceae_R-7_group* (*p* = 0.001), *Ruminococcus* (*p* < 0.001), *Ruminococcaceae_NK4A214_group* (*p* < 0.001), *UCG-004* (*p* = 0.002), *Saccharofermentans* (*p* < 0.001), *Prevotellaceae_UCG-003* (*p* = 0.021), *Lachnospiraceae_ND3007_group* (*p* = 0.014), *Oscillospira* (*p* < 0.001), and *Selenomonas* (*p* = 0.001).

We used the LEfSe method to better understand specific bacterial dominance in the two groups (Figure 5). o_Oscillospirales, f_Oscillospiraceae, c_Bacilli, *g_Oscillospira*, s_Oscillospira_guilliermondii, *g_UCG_004*, f_Erysipelatoclostridiaceae f_F082, o_ Erysipelotrichales, *g_Christensenellaiceae_R_7_group*, f_Christensenellaceae, o_Christensenellales, *g_Lachnospiraceae_ND3007_group*, f_Bacteroidales_BS11_gut_group, *g_NK4A214_group*, p_Verrucomicrobiota, f_Bacteroidales_RF16_group, o_WCHB1_41, c_Kiritimatiellae, *g_ Prevotelaceae_UCG_003*, *g_Papillibacter*, f_Eubacterium_coprostanoligenes_group, f_UCG_010, *g_Selenomonas*, and s_Selenomoras_ruminantium were very abundant in RT groups. s_rumen_bacterium_R_9, *g_Succinimonas*, f_Succinivibrionaceae, o_Enterobacterales, c_Gammaproteobacteria, and p_Proteobacteria dominated in the CON group.

### 3.5. Correlation Heatmap Analysis of Bacterial Communities

The interaction of rumen microbial communities with growth performance and blood metabolites in yaks was analyzed using a correlation heatmap. The findings revealed that transplanting rumen fluid altered the correlation between microbiota (Figure 6). In addition, we found that *Prevotellaceae_NK3B31_Group*, *SP3-e08*, *Lachnospiraceae_ND3007_group*, and *Treponema* correlated more strongly with growth performance, whereas *Fibrobacter* correlated more strongly with TG.

### 3.6. PICRUSt2 Function Prediction

To assess the functional characteristics of the yak rumen fluid flora before and after rumen fluid transplantation, we used PICRUSt2 to predict potential functions and compared the differences that existed before and after transplantation (Appendix A). The most abundant functional methods at the Kyoto Encyclopedia of Genes and Genomes 2 level were carbohydrate metabolism, replication and repair, translation, membrane transport, amino acid metabolism, nucleotide metabolism, energy metabolism, glycan biosynthesis and metabolism, metabolism of cofactors and vitamins, transport and catabolism, enzyme families, and folding. The relative abundance of sorting and degradation, signal transduction, and lipid metabolism were higher in the RT group. The RT group had a relative abundance of carbohydrate metabolism (*p* = 0.007), amino acid metabolism (*p* < 0.001), membrane transport, transport and catabolism, signal transduction, and lipid metabolism than those in the CON group. In contrast, the relative abundance of folding, sorting, and degradation (*p* = 0.045), replication and repair, translation, nucleotide metabolism, energy metabolism, glycan biosynthesis and metabolism, metabolism of cofactors and vitamins, and enzyme families in the RT group were all lower than those in the CON group.

## 4. Discussion

The successful transplantation of fecal microorganisms in mice revealed that metabolic disorders of the organism are closely related to microbial community composition and have been shown to restore gut function and animal health status, providing clues to microbiota study [12,13,23]. Similarly, rumen fluid transplantation in ruminants can reshape the rumen microbial community structure, thereby improving body health and production efficiency. Rumen fluid transplantation has been shown to positively affect intestinal development in pre-weaned calves [24]; however, transplantation did not significantly improve animal growth performance [25]. Therefore, this experiment was conducted to investigate the effect of rumen fluid transplantation on the rumen microbiota and the performance of yaks.

Ruminants possess a unique rumen microbiota that helps convert low-quality feed into high-quality protein and is associated with other product features, such as feed efficiency [26]. It was discovered [17] that periparturient cows achieved rumen microbial remodeling, significantly increased dry matter intake and feeding frequency, and improved rumen function after rumen fluid transplantation. In addition, ruminal contents interchanged in cows with different lactation efficiencies [19]. A study on the transplantation of rumen contents in beef cattle found [14] that the apparent digestibility of nutrients in recipient beef cattle was not affected; nevertheless, nitrogen digestibility was improved. These findings suggest that fresh rumen fluid transplantation can effectively improve the productive performance of recipient animals. Our study found that transplanting fresh rumen fluid to yaks had no significant effect on weight gain over 90 d. However, when there was no significant effect on weight gain, the monthly feed intake of the RT group was lower than that of the CON group, which improved yak feeding efficiency. Studies on goats found that transplantation of fresh rumen fluid also did not have a significant effect on weight gain and our study is consistent with this finding. Our study also showed that there was no significant difference in feed efficiency between the CON and RT groups of yaks at 1 and 2 months. However, in the 3 months, the RT yaks had significantly higher feed efficiency than the CON group. Moreover, there was a significant trend of increased feed efficiency in the RT group of yaks in the 3 months, which may be due to the rumen fluid transplantation accelerating the rumen microbial adaptation to the high-grain diet and improving the digestibility of the feed nutrients in the yaks.

Blood metabolites can reflect the health status of the animal and the digestion, absorption, and metabolism of nutrients in the diet [27]. For example, TP and GLU levels in animals can reflect the energy metabolism and nutritional status of the organism, whereas TG and CHO levels are important indicators of lipid metabolism [28]. The presence of TP content indicates high protein metabolism in the animal, which is beneficial to improving the immunity of the animal and promoting growth and development. Our study showed that on days 4 and 90, the Ca and TP contents of the RT group were significantly higher than those of the CON group. It is clear that rumen fluid transplantation facilitates the growth and development of the animals, which is consistent with the findings that the feed efficiency of yaks in the RT group was significantly higher than that in the CON group at month 3. TG is the most abundant lipid in animals, the main form of stored energy, and is negatively correlated with body fat utilization [29]. CHO is mainly synthesized in the liver and includes free cholesterol and cholesteryl esters. Our findings revealed that GLU, CHO, and TG levels were significantly higher in the RT group than in the CON group on day 28. This suggests that rumen fluid transplantation accelerated diet adaptation to some extent in yaks in the RT group and had a beneficial effect on the fat and sugar metabolism in the body, thus speeding up digestion and nutrient absorption. This was also consistent with the higher feed efficiency of yaks in the RT group in the absence of differences in daily weight gain between the two groups in month 2. The results of the heatmap analysis showed a significant positive correlation between Fibrobacter and TG, indicating that Fibrobacter can promote lipid metabolism and that the transplanted rumen fluid performed better for the RT group of yaks, regardless of the stage.

We studied the effect of rumen fluid transplantation on rumen flora in yaks using 16s rDNA high-throughput sequencing technology to show that rumen fluid transplantation changes the structure and composition of the rumen flora in yaks. A previous study reported that rumen fluid transplantation significantly increased the abundance and diversity of rumen bacterial communities in a sheep model of rumen acidosis [30]. In the present study, alpha and beta microbiota diversity indices were significantly different between the two groups, and RT significantly increased Shannon and Simpson indices in the CON yaks. Studies on PCoA findings reveal that the rumen bacterial community structure was not identical between CON and RT yaks on days 4 and 7 post-RT; nevertheless, after 7 d post-RT, samples from the two groups began to converge. Changes in the rumen community have been observed to occur mainly in the first week following the complete transfer of rumen content [31]. Our study is consistent with that of the present study, indicating that the rumen flora is remodeling. Overall, our findings revealed that yaks in the RT group had completed the process of rumen flora remodeling and reached a steady state by day 7. In contrast, it took at least 14 d for yaks in the CON group to attain the same steady state, and the RT group had a more diverse bacterial community. This also indicates that the rumen microbiota balance quickly responds to the disturbance of fresh rumen fluid and achieves the state of fresh rumen fluid, which is conducive to establishing a new balance in rumen microbiota.

At the phylum level, Bacteroida and Firmicutes were the dominant phyla of rumen microorganisms in yaks in our study, similar to the findings of previous studies [8,32], suggesting that these bacteria play a significant role in the rumen of ruminants. We observed an increase in the relative abundance of Firmicutes and a decrease in the relative abundance of Bacteroida, Proteobacteria, and Spirochaetes in the RT group, which is similar to the results of related studies [11]. The findings of additional comparison revealed that the differences showed dynamic changes and were detected mainly on days 7 or 14 after RT. Firmicutes were mainly responsible for the catabolism of fibrous material, whereas Bacteroidota was mainly responsible for the degradation of non-fibrous material [33,34]. The present study showed firmicutes significantly increased on days 3, 7, and 28 after RT. They significantly decreased after 60 d of RT, whereas Bacteroida significantly decreased at days 7 and 28 after RT and increased after 60 d of RT. This indicates that before 28 d after transplantation, yaks consumed more oat grass and less concentrated, whereas, after 60 d of transplantation, they consumed less crude and more concentrated food. Proteobacteria play an important role in biofilm formation, fermentation, and the digestion of soluble carbohydrates [35]. In the present study, the relative abundance of Proteobacteria was lower on days 3, 7, 14, and 28 after RT and higher at 60 and 90 d after RT, probably due to the higher crude protein level and lower fiber content in the concentrate feed consumed by yaks after 28 d of RT. Therefore, it was hypothesized that the relative abundance of Proteobacteria increased with higher protein levels in the feeds consumed by yaks after 60 d of RT. The relative abundance of Proteobacteria increased with an increase in protein level and was negatively correlated with the fiber level in the feeds. This also corresponds to the previous observation that yaks in the RT group had significantly higher feed efficiency in the third month than in the CON group.

At the genus level, *Prevotella*, *Rikenellaceae_RC9_gut_group*, and *Christensenellaceae_R-7_group* were dominant in the rumen. One of the important roles of *Prevotella* is to degrade nitrogenous nutrients such as protein in feed [36]. The relative abundance of *Prevotella* was lower on days 3, 7, 14, and 28 after RT; however, it was higher on days 60 and 90 after RT. This is because yaks consumed more concentrate feed after 60 d of RT, leading to higher crude protein levels. In our correlation heatmap analysis, we found a significant positive correlation between *Prevotellaceae_NK3B31_Group* and concentrate consumption, indicating a beneficial effect on animal growth and adaptation to concentrates. Compared to yaks in the CON group, the percentage of *Christensenellaceae_R-7_group*, *NK4A214_group*, *Prevotellaceae_UCG-004*, and *Prevotellaceae_UCG-003* increased after days 3, 7, 14, and 28 of RT; nevertheless, *Christensenellaceae_R-7_group* and *NK4A214_group* decreased, whereas *UCG-004* and *Prevotellaceae_UCG-003* increased, mainly occurring after 60 d of RT. The *NK4A214_group* belonging to Ruminococcaceae, rich in endo-1, 4-beta-xylanase, and cellulase genes, plays an important role in degrading cellulose and hemicellulose [37]. *Christensenellaceae _R-7_ group* and *NK4A214_group* both belong to Firmicutes [38] and mainly catabolize fibrous material, which may explain the increase in Firmicutes abundance in the RT group by 28 d and a decrease after 60 d. Additionally, the proportion of *Prevotellaceae_UCG-004* and *Prevotellaceae_UCG-003* in Bacteroidota decreased after RT60, which may explain the significant decrease in Bacteroidota abundance to a certain extent. We also found significant positive correlations between *SP3-e08*, *Lachnospiraceae_ND3007_group*, and *Treponema* and total feed intake, oat grass intake, and forage efficiency in our correlation heatmap analysis, suggesting that these genera have beneficial effects on growth performance.

Yaks have higher energy storage levels, lipid metabolism, glycan synthesis, and metabolic gene families than other breeds, according to suggestions deduced from functional predictions of yak rumen bacteria. Therefore, differences in these gene families may help yaks become more energy efficient. In this study, gene function prediction of the PICRUSt2 gene revealed that the gene function of the yak rumen flora was influenced by rumen fluid transplantation. At the secondary level of the Kyoto Encyclopedia of Genes and Genomes metabolic pathway, yak rumen flora genes were involved in metabolism-related pathways, such as carbohydrate metabolism, replication and repair, translation, membrane transport, and amino acid metabolism. Increased bacterial resistance, improved carbohydrate metabolism in vivo, and stable intestinal flora are all benefits of increased intestinal flora diversity. In addition, it has been found that amino acid metabolism promotes the accumulation of intracellular fatty acids, which not only provide energy to living organisms but also support cell membrane formation and alter cell membrane permeability [39]. In the present study, carbohydrate and amino acid metabolism were significantly higher in the RT group than in the CON group. Furthermore, the diversity of the flora increased after RT, indicating that RT contributes to carbohydrate and amino acid metabolism pathways in yak upregulation, which improves yak metabolism and effectively promotes yak growth. accelerating yak adaptation to optimal digestion of the served-feeding rations.

The findings demonstrate that rumen fluid transplantation may reshape the rumen flora structure and promote the rumen flora to adapt to house-feeding diets. Ruminal fluid transplantation improved yak feed efficiency in the RT group by rapidly reshaping the rumen flora and accelerating yak adaptation to digest the house-feeding rations.

## Figures and Tables

**Figure 1 microorganisms-11-01964-f001:**
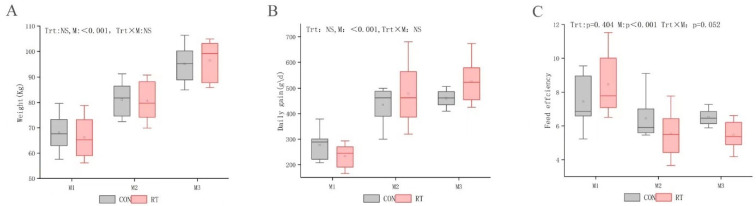
(**A**) monthly body weight (kg), (**B**) monthly daily weight gain (g), and (**C**) feed efficiency: amount of feed required to gain 1 kg feed efficiency of yaks in control (CON group, gray) and test (RT group, red) groups during the 3-month trial period.

**Figure 2 microorganisms-11-01964-f002:**
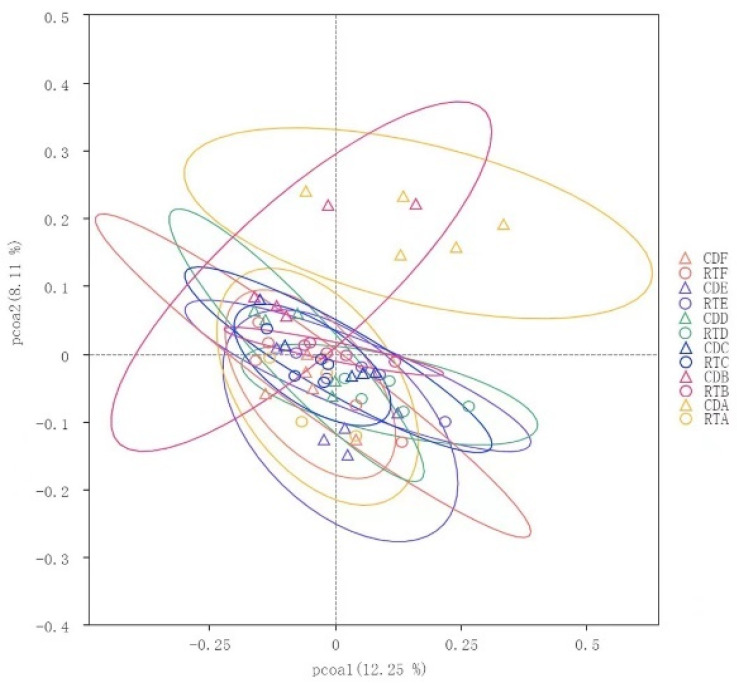
A, B, C, D, E, and F in control group (CON group) and experimental group (RT group) were represented by rumen microflora differences on days 4, 7, 14, 28, 60, and 90, respectively, calculated by weighted UniFrac distance and calculated by principal coordinate analysis (PCoA).

**Figure 3 microorganisms-11-01964-f003:**
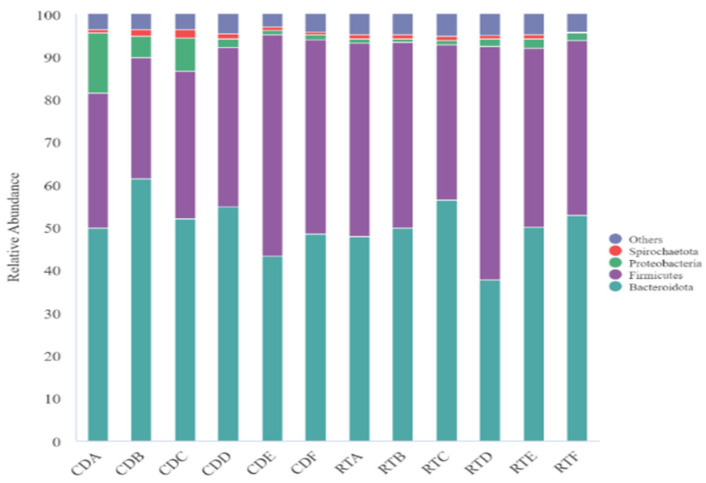
Distribution of rumen microbial communities at the gate in the control (CON group) and test (RT group) groups on days 4, 7, 14, 28, 60, and 90.

**Figure 4 microorganisms-11-01964-f004:**
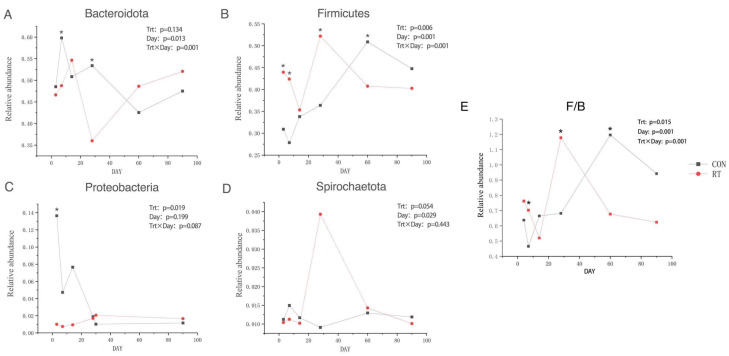
Dynamics of rumen fluid transplanted at the gate level on days 4, 7, 14, 28, 60, and 90 in control (CON group, blue) and test (RT group, red) groups (**A**) Phylum Bacteroida; (**B**) Phylum Thick-Walled Bacteria; (**C**) Phylum Aspergillus; (**D**) Spirochetes; (**E**) Firmicutes: Bacteroida ratio.

**Figure 5 microorganisms-11-01964-f005:**
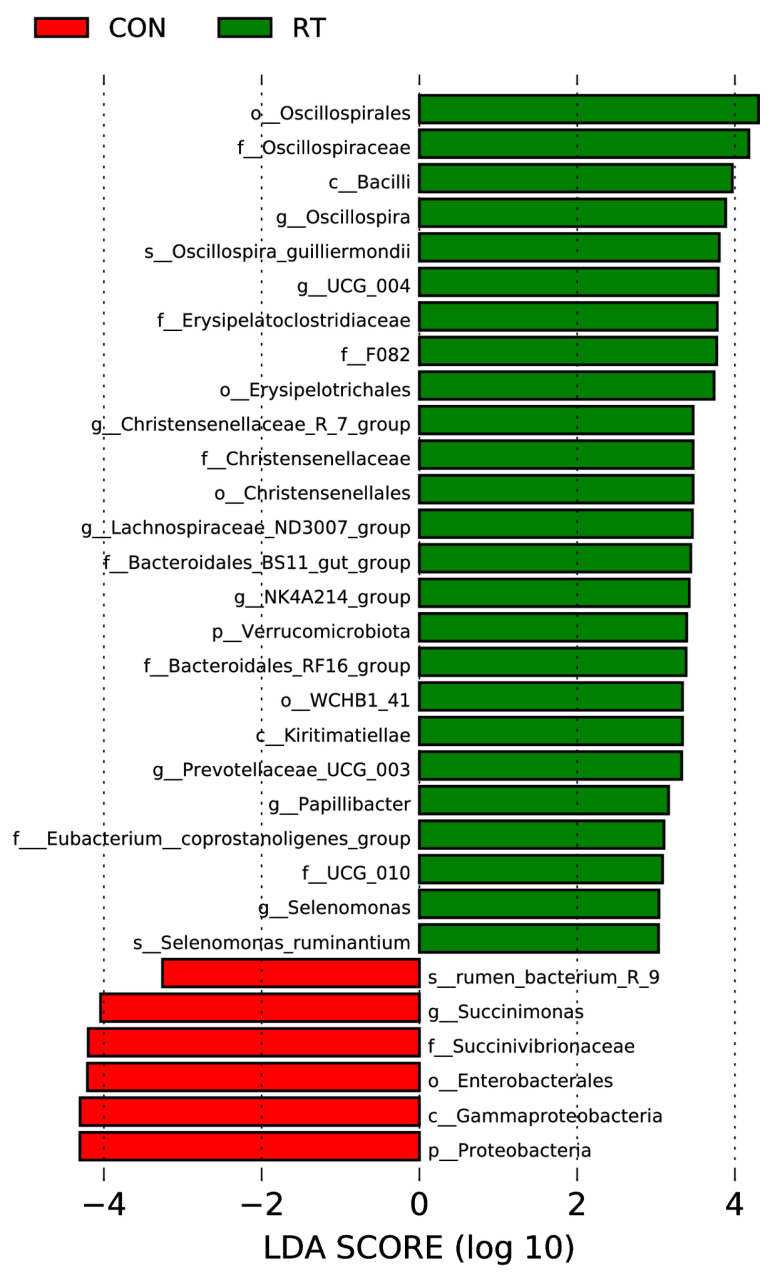
LEfSe analysis of the rumen microbiota of yaks in the control (CON) and test (RT) groups. Histogram of linear discriminant analysis scores based on categorical information.

**Figure 6 microorganisms-11-01964-f006:**
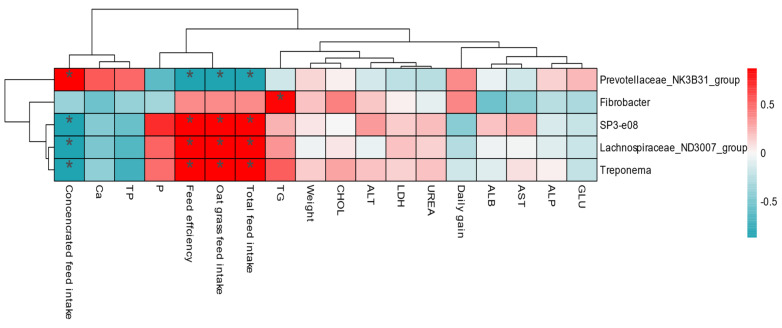
Correlation between rumen bacteria and growth performance. Each row in the graph represents a genus, each column represents a growth performance indicator, and each grid represents the Pearson correlation coefficient between a component and an indicator. Red represents positive correlations, while blue represents negative correlations. * indicates a 0.05 level of significance.

**Table 1 microorganisms-11-01964-t001:** Ingredients and nutritional composition of each diet.

Ingredients	Content (%)
Ingredients	
Com	50.00
Wheat	19.00
Rapeseed meal	6.00
Soybean meal	20.00
CaHPO4	1.00
Premix (1)	4.00
Total	100.00
Nutrient levels (2)	94.96
DM	15.85
CP	2.63
EE	44.53
NDF	25.18
ADF	0.87
Ca	0.46
P	

(1) The premix provided the following per kg of the concentrate supplement: Cu13 mg, Fe 75 mg, Zn 30 mg, Mn 35 mg, Se 0.2 mg, I 0.5 mg, Co 0.3 mg, VA 5 000 IU, VD 600 IU, VE 50 IU, non-starch polysaccharidase NSP enzyme 1000 U, and phytase 11,600 U. (2) Nutrient levels were all measured values.

**Table 2 microorganisms-11-01964-t002:** Effect of rumen fluid transplantation on blood metabolites of Yak.

Blood Metabolites	CON			RT			SEM	*p*-Value		
	Day 4	Day 28	Day 90	Day 4	Day 28	Day 90		Group	Day	Group*Day
ALB	30.79	30.4 ^b^	31.36	29.65	30.65 ^a^	31.31	2.11	0.62	0.32	0.63
ALP	40.75	146.38	193.63	94	143.25	170.38	64.93	0.47	0.00	0.04
ALT	23.5	26.25	31.88	24.88	30.5	30.88	9.71	0.58	0.12	0.75
AST	79.13	75.75	96.13	91	82	95	17.87	0.25	0.03	0.56
Ca	1.11 ^b^	2.22	2.29 ^b^	2.25 ^a^	2.29	2.37 ^a^	0.64	0.00	0.00	0.00
GLU	4.42	4.68 ^b^	5.28	4.65	5.23 ^a^	5.41	0.57	0.02	0.00	0.40
LDH	780.75	776.88	874.25	834.13	808.75	904.13	107.88	0.20	0.02	0.94
TP	56.25 ^b^	59.04	57.38 ^b^	58.27 ^a^	63.7	60.95 ^a^	3.99	0.00	0.00	0.53
UN	2.27	3.35	4.01	1.78	3.18	3.52	1.30	0.23	0.00	0.89
P	2.44	2.68	2.89	2.82	2.69	2.73	0.30	0.34	0.19	0.03
CHOL	1.84	1.65 ^b^	1.97	1.77	2.11 ^a^	2.3	0.38	0.01	0.02	0.07
TG	−0.02	0.06	0.21	0.02	0.12	0.22	0.12	0.10	0.00	0.75

ALB, albumin; ALP, alkaline phosphatase; ALT, alanine aminotransferase; AST, glutathione aminotransferase; Ca, calcium; GLU, glucose; LDH, lactate dehydrogenase; TP, total protein; UN, urea nitrogen; P, phosphorus; CHOL, cholesterol; TG, triglycerides. Peer data at the same time point with no letter or the same letter on the shoulder indicate a non-significant difference (*p* > 0.05) and different lowercase letters indicates a significant difference (*p* < 0.05). Group*Day is the interaction of the two factors.

**Table 3 microorganisms-11-01964-t003:** Effect of rumen fluid transplantation on VFA in yaks.

VFA	Treatment Groups												SEM	*p*-Value		
	CON						RT									
	Day 4	Day 7	Day 14	Day 28	Day 60	Day 90	Day 4	Day 7	Day 14	Day 28	Day 60	Day 90		Group	Day	Group*Day
Acetate	42.22	44.90	47.56	35.47	36.60	43.93	41.29	52.45	49.43	40.26	34.56	44.09	1.037	0.314	0.000	0.677
Propionate	12.33	16.26	17.34	9.74	9.05	9.49	10.72	13.79	14.13	10.02	8.21	11.65	1.038	0.239	0.000	0.424
Isobutyrate	0.52	0.41 ^b^	0.49	0.43 ^b^	0.40 ^b^	0.64	0.56	0.83 ^a^	0.64	0.73 ^a^	0.66 ^a^	0.73	1.038	0.000	0.246	0.055
Butyrate	7.12	7.40 ^b^	8.22	7.72	4.95	5.60	8.29	11.54 ^a^	9.44	6.49	6.98	8.57	0.781	0.005	0.013	0.171
Isovalerate	0.48	0.54	0.72	0.84	0.62	0.83	0.48	0.70	0.64	0.69	0.68	0.70	0.533	0.613	0.003	0.273
Valerate	0.30	0.40	0.40	0.33	0.15	0.24	0.30	0.46	0.41	0.22	0.26	0.44	0.473	0.266	0.019	0.352
Acetate Propionate	3.75	2.94	3.02	3.8	4.1	4.76	3.86	3.82	3.53	4.11	4.39	3.97	0.08	0.107	0.000	0.016

**Table 4 microorganisms-11-01964-t004:** Effect of rumen fluid transplantation on the alpha diversity of rumen flora in yaks.

Time	Group	Observed OTUs	Shannon Diversity	Simpson
Day 4	CON	1989 ± 215	7.1 ± 1.12	0.95 ± 0.06
	RT	2159 ± 130	8.94 ± 0.24	0.99 ± 0
Day 7	CON	1922 ± 249	7.47 ± 1.18	0.94 ± 0.08
	RT	2104 ± 138	8.8 ± 0.24	0.99 ± 0
Day 14	CON	1980 ± 150	7.98 ± 0.75	0.98 ± 0.02
	RT	2045 ± 130	8.54 ± 0.19	0.99 ± 0
Day 28	CON	1895 ± 135	8.11 ± 0.27	0.98 ± 0
	RT	2296 ± 176	8.86 ± 0.17	0.99 ± 0
Day 60	CON	1993 ± 69	8.38 ± 0.3	0.99 ± 0.01
	RT	2135 ± 100	8.58 ± 0.24	0.99 ± 0
Day 90	CON	1899 ± 89	8.45 ± 0.29	0.99 ± 0
	RT	1824 ± 156	7.97 ± 0.19	0.98 ± 0
*p*-value	Day	0.017	0.362	0.416
	Trt	<0.001	<0.001	0.013
	Trt × Day	0.034	0.001	0.181

Data are expressed as mean ± standard deviation. The Kruskal–Wallis rank sum test was used to test the statistical significance of treatment (Trt), day (Day), and day-by-day treatment interactions (Trt × Wk).

**Table 5 microorganisms-11-01964-t005:** Effect of rumen fluid transplantation on the level of rumen flora genus in yaks.

Genus	Relative Abundance, %												SEM	*p*-Value		
	CON						1.30									
	Day 4	Day 7	Day 14	Day 28	Day 60	Day 90	Day 4	Day 7	Day 14	Day 28	Day 60	Day 90		Group	Day	Group*Day
*Prevotella*	23.51	30.95	26.33	19.95	12.03	16.50	13.98	19.44	23.29	10.38	15.68	20.45	1.30	0.074	0.033	0.234
*Rikenellaceae* *_RC9_gut_group*	4.22	6.20	10.07	12.24	15.93	13.10	11.76 ^a^	11.29 ^a^	12.73	11.49	13.29	10.97	0.46	0.085	0.002	0.012
*Christensenellaceae* *_R-7_group*	1.54	1.58	1.97	2.33	3.75	3.30	3.01 ^a^	2.51 ^a^	2.33	3.76 ^a^	3.01	3.18	0.12	0.002	0.000	0.001
*Ruminococcus*	0.76	1.17	1.77	2.05	3.15 ^a^	2.70	2.19 ^a^	2.18 ^a^	1.60	2.20	1.93	2.69	0.10	0.069	0.000	0.000
*Ruminococcaceae* *_NK4A214_group*	0.74	0.79	1.51	1.83	2.81	2.43	2.15 ^a^	2.16 ^a^	1.78	2.66 ^a^	2.43	1.90	0.10	0.000	0.000	0.000
*Prevotellaceae* *_UCG-004*	0.38	0.83	1.28	1.33	1.45	1.15	2.39 ^a^	2.21 ^a^	4.04 ^a^	2.37	2.01	0.82	0.16	0.000	0.001	0.002
*Saccharofermentans*	0.95	1.13	1.60	1.75	2.38 ^a^	2.08 ^a^	1.93 ^a^	1.90 ^a^	1.42	1.99	1.63	1.39	0.07	0.568	0.014	0.000
*Prevotellaceae* *_UCG-003*	1.21	1.10	1.30	2.07	1.18	1.58	2.10 ^a^	2.04 ^a^	2.12 ^a^	1.55	2.01 ^a^	1.19	0.09	0.011	0.754	0.021
*Lachnospiraceae* *_ND3007_group*	0.16	0.44	0.85	2.17	2.64	2.14	1.91	3.80 ^a^	1.29	2.19	1.15	0.55	0.24	0.345	0.449	0.014
*Quinella*	0.46	0.91	2.77 ^a^	2.57	2.07	1.01	1.51	1.46	1.08	1.73	1.00	0.52	0.16	0.149	0.045	0.065
*Succiniclasticum*	1.51	0.67	1.23	1.74	1.25	1.62	1.04	1.00	0.82	1.75	2.05	2.20	0.13	0.585	0.113	0.625
*Oscillospira*	0.06	0.25	0.70	0.49	0.82	0.94 ^a^	1.65 ^a^	1.70	1.79 ^a^	4.26 ^a^	1.95 ^a^	0.47	0.17	0.000	0.000	0.000
*Prevotellaceae* *_UCG-001*	0.89	1.18	1.01	1.39 ^a^	1.31	1.13	0.98	1.20	1.20	0.53	0.81	1.12	0.08	0.279	0.939	0.391
*Succinimonas*	7.40	2.91	1.01	0.37	0.17	0.05	0.24 ^a^	0.05	0.19	0.09	0.03	0.14	0.61	0.132	0.458	0.495
*Selenomonas*	0.54	0.51	0.97	1.54 ^a^	0.97	1.08	1.19 ^a^	1.55	0.92	0.94	0.96	1.42	0.06	0.034	0.187	0.001

## Data Availability

The data that support the findings of this study are available from the corresponding author upon reasonable request, and the sequencing data are available from NCBI.

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
