# Peer review of "Ruminal Fluid Transplantation Accelerates Rumen Microbial Remodeling and Improves Feed Efficiency in Yaks"

_microorganisms, 2023, doi:10.3390/microorganisms11081964_

Round 1
Reviewer 1 Report
The article presents clearly the advantage of rumen fluid transplantation, a quite popular technique in ruminant medicine/management. I advocate for the publication of this work after some minor revisions:
- Line 132: 340 ml? please add the unit.
- Please use ml or mL
- Line 161-162: reformulated, describe the tube (EDTA, CAT coagulator factor)? Correct: 2 ml of the supernatant were stored in liquid nitrogen for xx days/hours?
- Line 166: CTAB add the complete term: cetyl trimethylammonium bromide
- Table 2. Please check the front of the letters of the superscript. (for example 5.23a: GLU in RT). Explain the superscript of all the tables.
- Line 295. “Significantly increased” with “s” not “S”.
- The Figure are very small. Please increase the size of the figures.
- Define the abbreviation in their first mention, not only in the Tables but also in the text, for example, GLU and GT (lines 232 and 233)
- Lines 369-370: I would suggest changing “fiber-rich-feed into high digestible bacterial protein”.
- Line 496: Ruminal (or rumen) fluidm nor "luminal"
- Line 499: please write: "accelerating yak adaptation to optimal digestion of the served-feeding rations."
the article is well written, some minor changes are required.
Author Response
Dear reviewer:
We are very grateful for your efforts in our manuscript and for giving us the opportunity to resubmit a revised version of our manuscript. Those comments are all valuable and very helpful for revising and improving our paper, as well as the important guiding significance to our research. We have studied comments carefully and have made corrections which we hope meet with approval. Revised portions are marked in red on the paper. The main corrections in the paper and the responses to the reviewer’s comments are as flowing:
- 1. Line 132: 340 ml? please add the unit. Please use ml or mL
Response1: We appreciate these comments. I have made changes in the text.
- Line 161-162: reformulated, describe the tube (EDTA, CAT coagulator factor)? Correct: 2 ml of the supernatant were stored in liquid nitrogen for xx days/hours?
Response2: We express our deepest gratitude for your careful work and thoughtful suggestions. I have made changes in the text.
- Line 166: CTAB add the complete term: cetyl trimethylammonium bromide
Response3: We appreciate these comments. I have noted in the text.
- Table 2. Please check the front of the letters of the superscript. (for example 5.23a: GLU in RT). Explain the superscript of all the tables.
Response4: We express our deepest gratitude for your careful work and thoughtful suggestions. I have noted in the text
- Line 295. “Significantly increased” with “s” not “S”.
Response5: We express our deepest gratitude for your careful work and thoughtful suggestions. I have made changes in the text.
- The Figure are very small. Please increase the size of the figures.
Response6: We are grateful for the comments. I have changed the size.
- Define the abbreviation in their first mention, not only in the Tables but also in the text, for example, GLU and GT (lines 232 and 233)
Response7: Thank you very much for your suggestion. I have explained this in Table 2 below.
- Lines 369-370: I would suggest changing “fiber-rich-feed into high digestible bacterial protein”.
Response8: We thank you for your questions about this study. I have not been able to find a corresponding question in the text.
- Line 496: Ruminal (or rumen) fluidm nor "luminal"
Response9: Thank you very much for your suggestion. I have made changes at line 503.
- Lines 499:please write: "accelerating yak adaptation to optimal digestion of the served-feeding rations."
Response10: Thank you for your suggestion. I have made changes to the text.。
We sincerely appreciate your hard work and thoughtful suggestions, which have greatly improved this article.
Reviewer 2 Report
This study focused on the changes in rumen flora due to dietary changes and the improvement of rumen fluid transplantation on the remodeling of ruminal flora of the yak and production performance when yak breeding patterns were changed. It is an interesting report and within the scope of the journal. However, a major revision is needed before reconsideration. Please address the appended comments:
Line 73: How long is this stage? Please add a reference.
Line 74-76: please add more information.
Line 90: adult animals. Please indicate the species of animals. Also, consider this comment in the whole manuscript.
Line 100-102. However, there has been little research into how rumen fluid trans- 100 plantation affects rumen fermentation and microbial community dynamics in yaks 101 adapted to high grain ratios. I recommend the authors to highlight the reason for this statement.
Line 113: s rRNA gene amplification and MiSeq sequencing should be explained well in the introduction section. Why the authors did this?
Line 115: Please add a short description of the animals used in this study (puberty, seasonality, economic importance,..)
Line 116-117: The Institutional Animal Care and Use Committee of Qinghai University approved all procedures in this study. Please add the ethical code or approval number for this study.
Line 149: All steps of the experiment should be illustrated in a figure. I recommend this.
Line 159-160 Why the authors selected these time points in the experimental procedures?
Line 165-180: Please add relevant references to this section.
Line 240, 242: Why the values in each column were not presented as mean± its own SEM
Line 270-273: This legend is not enough. Please add more description. Also in Figure 5.
Line 320-323: this figure is not clear. Please refer to the significant findings in this figure and all figures of the study.
Line 328-340: These findings should be indicated in the methodology section. Please consider this issue, especially in the statistics section (lines 203-220).
Line 328: please add the main findings in this section and try to illustrate these findings based on the measured parameters and previous literature.
Line 377: These findings refer to what?
Line 369-389: The authors should discuss the reasons for these findings, not compare them with other studies.
Line 412- studied not study
Line 478-488: relevant references should be indicated
Line 496: Please create a conclusion section. What are the shortcomings of this study?
Minor editing of English language required
Author Response
Dear reviewer:
We are very grateful for your efforts in our manuscript and for giving us the opportunity to resubmit a revised version of our manuscript. Those comments are all valuable and very helpful for revising and improving our paper, as well as the important guiding significance to our research. We have studied comments carefully and have made corrections which we hope meet with approval. Revised portions are marked in blue on the paper. The main corrections in the paper and the responses to the reviewer’s comments are as flowing:
- Line 73: How long is this stage? Please add a reference.
Response1:We express our deepest gratitude for your careful work and thoughtful suggestions. Based on practical experience, we found that the adaptation phase varied from 10-30 days, while anorexic behaviour was observed, so we did the present study.
- Line 74-76: please add more information.
Response2: We express our deepest gratitude for your careful work and thoughtful suggestions. We are very sorry that we are not able to provide more references for the first rumen fluid transplantation trial on yaks.
- Line 90: adult animals. Please indicate the species of animals. Also, consider this comment in the whole manuscript.
Response3: We appreciate these comments. I have modified the question in the text, as there are fewer relevant references for yaks, so we refer more to other ruminants.
- Line 100-102. However, there has been little research into how rumen fluid trans- 100 plantation affects rumen fermentation and microbial community dynamics in yaks 101 adapted to high grain ratios. I recommend the authors to highlight the reason for this statement.
Response4: We express our deepest gratitude for your careful work and thoughtful suggestions. The results of our team's study showed that concentrate feed affects the abundance and diversity of yak rumen flora, so we applied yak rumen fluid, which had been adapted to concentrate feed, to improve the rumen flora structure of grazing yaks.
- Line 113: s rRNA gene amplification and MiSeq sequencing should be explained well in the introduction section. Why the authors did this?
Response5: We express our deepest gratitude for your careful work and thoughtful suggestions. I have explained this in the Materials and Methods section of the text.
- Line 115: Please add a short description of the animals used in this study (puberty, seasonality, economic importance,..)
Response6: We are grateful for the comments. In this experiment, The experiment was carried out during the winter grass period, and the test yaks were all 6 months old and not in heat, which is the vast majority of the economic income of the local nomadic people.
- Line 159-160 Why the authors selected these time points in the experimental procedures?
Response7: We express our deepest gratitude for your careful work and thoughtful suggestions. These time points were selected with reference to dairy related trials.
- Line 240, 242: Why the values in each column were not presented as mean± its own SEM
Response8: Thank you very much for your suggestion. This expression was chosen because of the large number of indicators and time points measured.
- Line 412- studied not study
Response9: We express our deepest gratitude for your careful work and thoughtful suggestions. I have made changes in the text.